# The Chronic Effects of a Single Low-Intensity Blast Exposure on Phosphoproteome Networks and Cognitive Function Influenced by Mutant Tau Overexpression

**DOI:** 10.3390/ijms25063338

**Published:** 2024-03-15

**Authors:** Marcus Jackson, Shanyan Chen, Thao Thi Nguyen, Heather R. Siedhoff, Ashley Balderrama, Amitai Zuckerman, Runting Li, C. Michael Greenlief, Gregory Cole, Sally A. Frautschy, Jiankun Cui, Zezong Gu

**Affiliations:** 1Department of Pathology and Anatomical Sciences, University of Missouri School of Medicine, Columbia, MO 65212, USA; mnjdgx@umsystem.edu (M.J.); chenshan@health.missouri.edu (S.C.); h.r.siedhoff@wustl.edu (H.R.S.); abhyk@missouri.edu (A.B.); amitai.zuckerman@gmail.com (A.Z.); lirun@health.missouri.edu (R.L.); cuij@missouri.edu (J.C.); 2Charles W. Gehrke Proteomics Center, University of Missouri, Columbia, MO 65211, USA; tnqp6@missouri.edu (T.T.N.); greenliefm@missouri.edu (C.M.G.); 3Harry S. Truman Memorial Veterans’ Hospital Research Service, Columbia, MO 65201, USA; 4Geriatric Research Education and Clinical Center (GRECC), West Los Angeles Healthcare System, Los Angeles, CA 90073, USA; gregorycole.ucla@gmail.com (G.C.); sfrautschy@mednet.ucla.edu (S.A.F.); 5Departments of Neurology and Medicine, David Geffen School of Medicine, University of California, Los Angeles, CA 90095, USA

**Keywords:** low-intensity open-field blast, discrimination-learning index, phosphopeptides, label-free DIA-PASEF quantitative proteomics, phosphoproteomic profiling, WpCNA, eigenpeptides

## Abstract

Blast-induced neurotrauma (BINT) is a pressing concern for veterans and civilians exposed to explosive devices. Affected personnel may have increased risk for long-term cognitive decline and developing tauopathies including Alzheimer’s disease-related disorders (ADRD) or frontal-temporal dementia (FTD). The goal of this study was to identify the effect of BINT on molecular networks and their modulation by mutant tau in transgenic (Tg) mice overexpressing the human tau P301L mutation (rTg4510) linked to FTD or non-carriers. The primary focus was on the phosphoproteome because of the prominent role of hyperphosphorylation in neurological disorders. Discrimination learning was assessed following injury in the subsequent 6 weeks, using the automated home-cage monitoring CognitionWall platform. At 40 days post injury, label-free phosphoproteomics was used to evaluate molecular networks in the frontal cortex of mice. Utilizing a weighted peptide co-expression network analysis (WpCNA) approach, we identified phosphopeptide networks tied to associative learning and mossy-fiber pathways and those which predicted learning outcomes. Phosphorylation levels in these networks were inversely related to learning and linked to synaptic dysfunction, cognitive decline, and dementia including Atp6v1a and Itsn1. Low-intensity blast (LIB) selectively increased pSer262tau in rTg4510, a site implicated in initiating tauopathy. Additionally, individual and group level analyses identified the Arhgap33 phosphopeptide as an indicator of BINT-induced cognitive impairment predominantly in rTg4510 mice. This study unveils novel interactions between ADRD genetic susceptibility, BINT, and cognitive decline, thus identifying dysregulated pathways as targets in potential precision-medicine focused therapeutics to alleviate the disease burden among those affected by BINT.

## 1. Introduction

### 1.1. Long Term Effects of Blast-Induced Neurotrauma

Primary blast-induced neurotrauma (BINT) remains a significant health concern for veterans and servicemen during training operations and deployment missions, increasing risk for neuropsychiatric disorders such as post-traumatic stress disorder (PTSD) and/or neurodegenerative diseases [1,2]. Primary BINT due to low-intensity blast (LIB) is caused by the transmission of hypersonic shockwave energy [3,4] passing through the skull and meningeal layers [5], then penetrating brain parenchyma, causing phonon decays and/or stretch/strain in the neurovasculature [6,7], which contributes to a complex set of neuropsychological sequalae [1,8,9,10].

### 1.2. Utility of Phosphoproteomics

In BINT, there is emerging evidence of global proteome dysregulation impacting excitotoxicity [11], and hyperphosphorylation of tau [12,13]. However, there have been a limited number of studies on phosphoproteomics in BINT [12,14]. Recent innovations in mass spectrometry (MS) technology, such as the incorporation of collisional cross section values [15,16], data-independent acquisition (DIA) [17], parallel accumulation-serial fragmentation (PASEF) [16,18], label-free technology, and titanium dioxide enrichment [19], have led to increased coverage of the identified LIB-exposed animal global and phosphoproteomes [7,11], uncovering brain signaling mechanisms related to BINT pathophysiology [12]. In our previous work, BINT-induced differential changes to phosphoprotein expression were most related to learning, axon guidance, and synaptic plasticity pathways; however, few studies have investigated the impact of BINT on mice phosphoproteomes. Although we identified hyperphosphorylation of the tau protein following BINT, it is unknown whether these phosphorylation events occur at tau sites distinct from other mechanisms of TBI, and how BINT alters the mutant tau interactome.

### 1.3. Linking Tau Hyperphosphorylation with Complex Behavioral Phenotypes and Synaptic Pathology Following BINT

Tau pathology is a hallmark of brain injury associated pathogenesis. Transgenic human tau mouse models such as the rTg4510 mice exhibit premature (≤7 months) tau histopathology along the anterior-posterior (A-P) axis of the mouse brain with 20% forebrain loss by 7 months of age and accelerated tau pathology [20,21,22,23,24] with behavioral deficits such as grooming, abnormal gait, loss of righting reflex, touch response, hanging behavior, forelimb placing reflex, and hyperactivity, typically noticeable as early as 4 months of age [22,24] or learning deficits even as early as 10 weeks of age [25]. TBI can increase tau oligomers and induce cognitive deficits in human tau Tg mice [26], which can be reversed by administering doxycycline [27]. Thus, LIB-induced phosphoproteome alterations in rTg4510 mice might reveal pathways involved in the interaction between BINT, tau aggregation, and cognitive deficits.

In mouse brains, the tau protein is increased at hyperacute (3 h) and acute phases (24 h) but not at the chronic phase (30 DPI) following a single blast exposure [12]. As we previously described, BINT-induced tau hyperphosphorylation increased at 7- and 30-DPI in the cortex of C57Bl/6J mice [13]. Ultrastructural analysis showed that these mice also displayed reduced cortical synapses and active zone length at these same time points, owing to a possible link between LIB-induced increased tau expression and nanoscale synaptic dysfunction [11,13].

Although the tau protein is predominantly axonal, dendritic tau plays a crucial role in mediating synaptic processes including exocytosis and excitability amongst others [13,28]. For example, Dr. Li Gan and colleagues unveiled the tau interactome in humans suffering from TBI, which provided insights into the potential tau-dependent mechanisms related to synaptic dysfunction [28]. They proposed that following injury, the C-terminus of tau directly binds with munc-interacting proteins (mint), mediating munc18-SNARE protein interactions, and thereby affecting synaptic vesicle exocytosis [28]. In its hyperphosphorylated, pathological state, tau is transported to synapses and has been identified in human synapses, especially in Alzheimer’s disease (AD) cases [29,30,31]. At the presynaptic level, kinases such as CaMKII, CDK-5, and GSK-3β can help induce tau oligomerization mediated by 14-3-3 adapter proteins which can modulate synaptic tau clearance mechanisms with possible deleterious effects on SNARE protein expression and synaptic mitochondria bioenergetics [28,29]. Additionally, in mice overexpressing the phosphorylated tau (p-tau) protein in the brain, PSD-95 protein expression is increased along with glutamate-dependent neural hyperexcitability [11,13].

### 1.4. Knowledge Gaps in Understanding the Role of Tau in Risk of BINT

The genetic and environmental risk factors predisposing certain individuals to neurological disease in response to BINT are not well understood. Low-intensity blast (LIB, <100 kPa) incidents, common in training operations, can induce concussive or mTBI-like symptoms such as headache, tinnitus, sensorineural hearing loss, memory loss, mood dysregulation, oculomotor symptoms, and cognitive dysfunction. Most personnel recover within 6–12 months of LIB exposure [10]. Although a clear pathological link between a single LIB exposure and chronic neurological disease risk such as dementia has not been established, repetitive LIB can cause cumulative shear-strain effects on brain cells [3] resulting in increased expression of dementia-related biomarkers such as amyloid β [32], hyperphosphorylated Tau (p-Tau) [33], neurofilament light chain, and many others [34]. Moreover, emerging evidence has found blast-induced impairment of perivasculature in the cerebellum, which also helps control complex cognitive functions besides its role in movement [35]. Finally, accumulation of hyperphosphorylated tau aggregates can also be central to symptoms and pathognomic. For example, perivascular and periventricular tau accumulation is a hallmark of chronic traumatic encephalopathy (CTE) [36], while more selective laminar distribution in the frontal and temporal cortices are hallmarks of frontal-temporal dementia (FTD) [37,38]. The Alzheimer’s Disease Cooperative Study Prevention Instrument Project found that modified Mini-Mental State Examination, anxiety, depression, irritability, and apathy levels at baseline were significant predictors of AD diagnosis [39]. These insights illuminate the importance of contextualizing BINT symptoms with individual-level molecular (e.g., genomic, transcriptomic, proteomic, and metabolomic) profiles to identify molecular drivers of BINT-related clinical sequelae that pose significant risk for future development of ADRDs.

### 1.5. Unveiling the Relationship between ADRD Genetic Markers and LIB-Induced Behavioral Deficits

Our recent work comparing the position-dependent blast effects across brain regions found the cortex proteome to be most affected relative to other regions in upright and prone-positioned animals; however, effects on the proteome in the upright-positioned animals were seen in synaptic, mitochondrial, and metabolic functional pathways [40]. Cortex pathology following BINT has also been indirectly linked with tauopathy and complex behavioral deficits in animal models [12]. We previously found that cortex asymmetric synapse count was reduced in BINT mice [13], which have been associated with poor cognition and learning abilities [41,42,43]. These mice were found to have hyperphosphorylation of the tau protein in the cortex [13]. Other behavioral deficits associated with BINT in rodents include anxiety-like behaviors [44], poor cognition and cognitive flexibility [11], learning deficits [11], reduced motor and sensorimotor function [12], weight loss [45], and apnea [45], many of which have been observed in rTg4510 mice [22,46]. Three months following BINT, mice displayed reduced discriminatory learning and reversal learning abilities which correlated with protein expression changes related to synaptic plasticity in the hippocampus [11]. There is, however, a paucity of studies that evaluate the dynamic molecular events in the cortex and how these events influence behavior.

### 1.6. Identification of Phosphopeptide Co-Expression Networks Related to Cognitive Changes

In this study, we sought to identify differentially changed phosphopeptides that can explain the LIB and tau effects on learning behavior in mice following a single LIB exposure at both individual and group levels. At the individual level, the phosphoproteome expression profiles of mice with the poorest learning abilities will be compared to other animals to identify distinct molecular responses to LIB that may confer learning deficits. At the group level, we will compare the expression of learning-related phosphopeptides across different LIB vs. sham comparison groups, which will unveil phosphopeptide candidate biomarkers that may underly tau-blast interaction effects on learning abilities. Functional characterization will be performed on these phosphopeptides to identify targetable brain pathways that are altered molecular responses following LIB may underly poor learning.

## 2. Results

### 2.1. Quantitative Proteomics and Behavior Prediction by Ingenuity Pathway Analysis (IPA)

Of 17,637 phosphopeptides identified via MS^2^ (Appendix A—MS^2^ raw data, MS^2^ raw data_rTg4510, and MS^2^ raw data_non-carriers), 706 and 550 were significantly changed in rTg4510 and non-carrier LIB-exposed mice relative to unexposed sham controls, respectively. Among these changes, 360 and 367 phosphopeptides were decreased and 346 and 183 phosphopeptides were increased in rTg4510 (Figure 1A, Appendix A—MS^2^ significant phosphopeptides_rTg4510) and non-carrier (Figure 1A, Appendix A—MS^2^ significant phosphopeptides_non-carriers) LIB-exposed mice relative to unexposed sham controls (genotype-matched), respectively. As expected, there was very low overlap (2.28%) between the rTg4510 and non-carrier LIB-exposed mice phosphopeptidomes (Figure 1B). This is likely a result from the difference in genetic makeup between the rTg4510 and non-carrier mice and their subsequent response to LIB. Using machine learning (ML) driven capabilities in IPA (RRID: SCR_008653; QIAGEN Ingenuity Pathway Analysis; Germantown, MD, USA), learning was predicted to be reduced in LIB-exposed non-carrier and rTg4510 mice aged 14 weeks old (Figure 1C). It should be noted that these predictions are inferred and should not be interpreted as definitive due to the large influence of the phosphopeptide expression sign (Log_2_ fold-change, ±). Intriguingly, the tau pSer262 site was selectively hyperphosphorylated in rTg4510 LIB-exposed mice relative to unexposed sham controls (Appendix A—MS^2^ significant phosphopeptides_rTg4510) whereas many other sites were reduced in non-carriers (Appendix A—MS^2^ significant phosphopeptides_non-carriers).

### 2.2. Results from Automated Home-Cage Monitoring System on the CognitionWall Experiment

Discrimination learning ability in 20 mice was assessed 30 days following a single LIB in an aHCM CognitionWall platform environment. Thirty days post LIB, during a 48-h period, mice were tracked via aHCM video recording for correct entries (left entrance), incorrect entries (middle or right entrances), and total entries (any entrance) (Figure 2A). These parameters were selected amongst hundreds of the dynamic measurement within the CognitionWall assessment to estimate the learning index, defined as the learning growth over a 48-h period (Figure 2B; Appendix A—Cogwall data) (Equation (3)). The learning index (continuous variable) was transformed into a learning level (categorical variable; scale 1 to 5) on a 5-point scale (5 = fast learning, 4 = normal learning, 3 = mildly slow learning, 2 = moderately slowed learning, and 1 = severely slowed learning) where each category corresponds to a number of standard deviation(s) away from the mean learning index (Mean learning index: 0.40) (Figure 2A). In this study, blast-exposed rTg4510 mice demonstrated a lower learning index than all other groups (Figure 1B), with the lowest improvement in learning from day one to day two observed in the rTg4510 4 animal (Figure 2B,C; red solid and red dashed lines). These findings were supported by average-linkage clustering which identified the blast rTg4510 4 animal as an outlier (Figure 2C).

Although bioinformatic predictions of blast vs. sham (genotype controlled) mouse learning performances at the group level was inconsistent with real learning performances at the group level (Figure 1C and Figure 2B), WpCNA assessed whether there are any individual-level phosphopeptidomic distinctions (e.g., module eigenpeptide values) that may confer poor learning in the blast LIB groups or notably in the rTg4510 4 animal with more serious impairment.

### 2.3. Weighted Peptide Co-Expression Network Analysis (WpCNA): Network Construction and Topological Overlap Matrix

A total of 17,637 phosphopeptides identified by MS^2^ in 20 animals were imported into RStudio as the expression data frame, and two columns into a .csv file containing animal identifiers and learning index (continuous variable) were imported as a functional outcome data frame for WpCNA analysis (See RMarkdown code for entire analysis in Appendix A—WpCNA code in RMarkdown). The continuous form of learning index was chosen over the categorical learning scale due to greater statistical power. Prior to conducting WpCNA, it is necessary to choose a β (Power parameter) that satisfies the scale-free topology criterion [47,48]. We found that setting β to 8 was the only value of β that produced a scale-free network shown by the right-skewed histogram plot (Figure 3A, left panel) and the high model strength (Figure 3A, right panel) for Log_10_(p(k)) (Probability of connectivity) vs. βLog_10_(k) (Connectivity), which is derived from Equations (4)–(6), (Log10(A_k_) vs. βlog10(s_k_)).

Standard peptide screening (SPS) (analogous to standard gene screening) was introduced to filter the large phosphopeptide expression data frame of 17,637 phosphopeptides to 564 phosphopeptides based on their marginal linear correlation coefficient (r) with learning index for all 20 animals (Threshold: r coefficient ≥ 0.05) (Appendix A—Standard Peptide screening results) [49]. The next step was to reduce our filtered multidimensional (≥3 expression columns) expression dataset of 564 phosphopeptides for each animal (n = 20) into modules that summarize the expression of 564 phosphopeptides from each animal into a single value (signed), the module eigenpeptide (analogous to module eigengene) [50]. To identify modules, a hierarchical clustering tree based on the weighted topological overlap matrix between phosphopeptides was plotted directly over a color map where the tips of the ”tree-branches” correspond to modules (Figure 3B) [51]. The height (y-axis) in which the tree is cut horizontally will remove upstream branches and phosphopeptides, while sparing downstream branches [51]. Three different tree cutting methods (labels adjacent to colormap) were compared to one another to determine which method successfully stratified most of the tree tips into distinct modules (denoted as colors), indicated by the color map in Figure 3B (directly below cluster tree) [51]. The colorstatic topological overlap matrix (TOM) tree-cutting method (first colormap in Figure 1B) obtains few and large clusters based on the characteristic pattern of fluctuations between the phosphopeptide pairs’ height (y-axis in Figure 3B) [51]. The colorDynamic TOM tree-cutting method first obtains a few small clusters from the colorstatic TOM method, then it splits the large clusters into subclusters by searching for patterns of fluctuations between the cluster tips [51]. This approach considers information from the dendrogram alone. Conversely, the colorDynamicHybrid TOM tree-cutting method defines clusters by their proximity to “objects” [51]. The colorDynamic TOM method was chosen as the preferred tree-cutting method because the largest cluster tips were separated into different modules (see module assignment in Appendix A—Phospep module assignment). Following network construction, network topology was computed based on the TOM dissimilarity (TOM similarity is not an acceptable input) which is defined as: 1−TOMsimilarity (Topological overlap) (Equation (7)) [47]. The turquoise and black modules had the greatest topological overlap indicating these modules contain highly interconnected phosphopeptides (Figure 3C). Additionally, there was high intermodular topological overlap between the turquoise and black modules indicating the weighted co-expression of their phosphopeptides are high.

#### 2.3.1. Weighted Peptide Co-Expression Network Analysis: Module–Module (Clustering and TOM Dissimilarity) and Module–Learning (Pearson Correlation) Relationship

Module eigenpeptides (analogous to the module eigengene) represent the first principal component of the expression (phosphopeptide Log_2_ intensity) data frame and summarize the multidimensional expression profile into a single dimensional value (signed or unsigned). Module eigenepeptides below 1 (blue boxes in Figure 4A) or above 1 (red boxes in Figure 4A) indicate that the peptides in a given module, for a given animal, are either decreased or increased together.

Nine module eigenpeptides across 20 animals (i.e., a total of 180 module eigenpeptides) were detected using the colorDynamic TOM tree-cut method, complete-linkage clustering, and Euclidian distance method, which are shown in Figure 4A (See module eigenpeptides in Appendix A—Module eigenpeptides for each animal). Module eigenpeptide dissimilarity, defined as 1—weighted correlation between module eigenpeptide in animal 1 and module eigenpeptide in animal 2 (iterated over all possible animal pairs), were used to plot a cluster dendrogram (Figure 4B) where turquoise and black module–module linkage (i.e., high correlation between module eigenpeptides) can be seen, amongst others.

In this study, the module significance (i.e., average of peptide significance for learning index across 20 animals) was greatest in the linked turquoise and black modules (Figure 4B). These modules may contain candidate phosphopeptides with expression patterns that represent phosphopeptidomic differences between animals (Figure 4A) and possibly BINT sequelae following single LIB exposure.

These findings prompted us to inquire further into the significantly low turquoise and black module eigenpeptides for the rTg4510 4 LIB-exposed animal (Figure 4A). This animal also displayed the lowest learning index of all animals. The phosphopeptide intensities in all 129 and 21 phosphopeptides in the turquoise and black modules, respectively, were plotted for the rTg4510 4 animal and were compared with the average intensities of non-carrier (n = 4) and rTg4510 unexposed sham controls (n = 5), and the remainder of the rTg4510 animal group (n = 4) (Figure 4C). Larger distances of phosphopeptide intensities between the rTg4510 4 blast animal and controls were identified as potential candidates that may explain the poor learning performance in the rTg4510 4 LIB-exposed animal.

#### 2.3.2. Weighted Peptide Co-Expression Network Analysis: Phosphopeptide Module Connectivity Measure, Multidimensional Scaling, and Relationship with Learning Index

In order to elucidate the relationship between phosphopeptide expression profiles and module assignment, a module eigenpeptide-based connectivity measure referred to as the module membership (MM) was computed [48]. The MM was defined as the correlation coefficient between the phosphopeptide Log_2_ intensities and the module eigenpeptide across all 20 animals (Equation (1)):(1)$$\left(\mathrm{MM}\right)\;\mathrm{kMEmodule}\left(i\right)=\;\mathrm{cor}\left(x_{i},\mathrm{MEmodule}\right),$$
where x_i_ is the phosphopeptide expression profile of peptide i and MEmodule is the module eigenpeptide of a selected module [48].

This implies that the intramodular phosphopeptide hubs (hubs have the highest k connectivity) in turquoise and black modules tend to also display high correlation with the module eigenpeptide. It was also discovered that turquoise and black PS with learning index was highly correlated with the k connectivity (Figure 5B).

Classical multi-dimensional scaling (MDS) was performed based on the TOMdissimilarity to visualize module structure, particularly with special interest in the turquoise and black modules (using standard R function cmdscale) [48]. Classical MDS is analogous to principal component analysis with the exception that in MDS, the distance between nodes (phosphopeptides i and j) on a cartesian plane are computed, whereas in PCA, the correlation between two nodes (x_i_ and x_j_) are computed. It was observed that in the first two scaling dimensions (analogous to principal components), intramodular hub (fingertips in Figure 5C) distances in the turquoise and black modules were mostly superimposed with a few intramodular hubs in the turquoise module displaying extreme topological overlap (Figure 5C).

To identify phosphopeptides in our filtered expression dataset (564 phosphopeptides) that display high correlation with learning index based on MM (module-eigenpeptide-based connectivity measure) and PS (Pearson correlation with learning index), we performed network screening (NS). NS utilizes both standard and network methods to relate expression of a gene, protein, or phosphopeptide to a trait (e.g., learning index). It was discovered that the top 30 phosphopeptides that displayed high correlation with learning index were mostly (20/30 phosphopeptides) assigned to the turquoise (Rubcn, Itsn1, Pcm1, Dlgap2, Bcl11b, Eif5b, Speg, Rims2, Palm3, Pgrmc1, Sytl5, Atp1a2, Eif5, Camkv, Ctnna1, Nlgn3, Cyb5b) and black modules (Atxn2, Prkar2a, Cobl) (Figure 5D) while the other 10 phosphopeptides, were assigned to modules with low module significance with learning (See NS results in Appendix A—Network Screening).

### 2.4. Weighted Peptide Co-Expression Network Analysis: Functional Characterization of Turquoise and Black Learning-Related Modules

To functionally characterize the learning-related turquoise and black modules, gene ontology (GO) enrichment analysis within the clusterProfiler package (RRID: SCR_016884; version 3.0.4) of each module, separately, was performed (Figure 6A,B; See results in Appendix A—Black and turquoise module GO). The top biological processes (BP) were associative learning (Bonferroni corrected *p* value = 5.60 × 10^−5^) and protein import (Bonferroni corrected *p* value = 8.36 × 10^−3^) for turquoise and black modules, respectively. The top cellular components (CC) were myelin sheath and hippocampal mossy fiber to CA3 synapse for turquoise (Bonferroni corrected *p* value = 7.49 × 10^−12^) and black (Bonferroni corrected *p* value = 4.6 × 10^−2^) modules, respectively. The molecular functions that were the most enriched in the turquoise module were scaffold protein binding (Bonferroni corrected *p* value = 1.63 × 10^−6^), signaling adaptor activity (Bonferroni corrected *p* value = 1.64 × 10^−5^), and molecular adaptor activity (Bonferroni corrected *p* value = 1.72 × 10^−5^).

### 2.5. Post-Weighted Peptide Co-Expression Network Analysis: Receiver Operating Characteristic Analysis

To determine if modules can predict learning level (Categorical dependent variable) based on module eigenpeptides across all 20 animals to explain rTg4510 transgene and blast effects, we performed receiver-operating characteristics (ROC) analysis. It was discovered that the turquoise and black modules, defined by associative learning and hippocampal mossy fiber to CA3 synapse, were most predictive of learning level given the genetic differences between the mice.

The positive state for the learning level was set to 4 (≥mean learning index, Figure 7) for ROC prediction. Under the nonparametric assumption, the area under the curve (AUC; test accuracy) for the turquoise and black modules (Null AUC = 0.5) were 0.726 (Asymptotic 95% Confidence Interval = 0.498–0.954; Asymptotic *p* value = 0.052) and 0.619 (Asymptotic 95% Confidence Interval = 0.312–0.926; Asymptotic *p* value = 0.447), respectively. The maximum value of the Kolmogorov-Smirnov (K-S) metric (to assess model effectiveness) were 0.500 and 0.476 for turquoise and black modules, respectively. The overall model quality for turquoise and black modules were 0.50 and 0.31, respectively. The remainder of modules displayed poor model accuracy, significance, and model effectiveness (K-S).

### 2.6. Individual Level Analysis Identifies Phosphopeptides in Modules That Confer an Increased Risk for Learning Deficits Following a Single LIB Exposure

Thus far, the turquoise (associative learning) and black (hippocampal mossy fiber pathway) modules appear to be most predictive of learning level across all 20 LIB-exposed animals. Given that our conditions are mild (46.6 kPa static overpressure), it is not surprising that few animals displayed low learning levels. We thereby sought to identify the candidate phosphopeptides related to these modules whose expression may confer an increased risk for learning deficits following a single LIB exposure, in subacute phase following exposure for the rTg4510 4 animal relative to other animals (Figure 8A). This animal’s enhanced behavioral response to LIB exposure was not due to blast experiment setting differences (Appendix A); thus, we postulate that molecular response differences may be involved. The cumulative distance was computed using the following equation (Equation (2)):(2)$$\mathrm{Cumulative}\;\mathrm{Distance}=\left(I_{0}-{\mathrm{avgI}}_{\mathrm{bt}}\right)+\left(I_{0}-{\mathrm{avgI}}_{\mathrm{st}}\right)+\left(I_{0}-{\mathrm{avgI}}_{\mathrm{sn}}\right),$$
where I_0_ is the intensity of a given phosphopeptide in the rTg4510 4 animal, I_bt_ is the average intensity of the phosphopeptide in the rTg4510 LIB-exposed group (n = 5), I_st_ is the average intensity of the phosphopeptide in the rTg4510 unexposed sham group (n = 4), and I_sn_ is the average intensity of the phosphopeptide in the non-carrier unexposed sham group (n = 6).

A dot plot was constructed to visualize the cumulative distance in the turquoise and black modules where the most changed phosphopeptides are highlighted (Figure 8B). The top ten phosphopeptides were considered candidates that may explain the blast rTg4510 4 animal’s poor learning performance. To visualize the co-expression interconnectivity (topological overlap) of the top ten candidates in relation to the entire phosphopeptide network, a circularized TOM similarity plot was constructed where candidate phosphopeptide interconnectivities with all module phosphopeptides can be seen (Figure 8C). Nodes correspond to individual phosphopeptides and edges represent the topological overlap between two connected phosphopeptides with an edge threshold of >0.5 (i.e., 1 is maximum and 0 is minimum topological overlap). Five of the ten phosphopeptides (Mapk8ip3, Gab1, Arhgap33, Gap43, and Slc1a2) in the turquoise module were increased in the blast rTg4510 4 animal relative to comparison groups, four were increased (Otub1, Elmo1, Ccm2, and Trip12), and one phosphopeptide in the black module was decreased (Bcl2l13) 40 days following a single LIB exposure. The most enriched GO term for each candidate was plotted as well. These signaling pathways may underly the link between LIB exposure and subsequent learning deficits in P301L mutant tau protein overexpressing rodents.

### 2.7. Group Level Analysis Using Protein Prioritization Identifies Blast-Relevant Phosphopeptides Related to Abnormal Behavorial and Molecular Phenotypes

One-hundred and fifty phosphopeptides in the turquoise and black modules were ranked (from 1 to 150) across five different comparison groups (or rank sets) by log_2_ fold change and −log_10_ *p*-value in each rank set (Figure 9A). The integrative rankings for each phosphopeptide were computed based on the sum of all their individual ranks in for each rank set (i.e., 5 rank sets × 2 ranks per rank set = 10 ranks). The top 40 integrative ranks were imported into STRING for peptide–peptide functional network construction and annotation to identify the most enriched mammalian phenotypes (using Mammalian Phenotype Ontology (MPO)). These 40 identified phosphopeptides were among the turquoise module, which is unsurprising given its size and predictive capabilities of cognitive performance. Among these, the Arhgap33 phosphopeptide was the top ranked, and deemed the most blast-relevant (Figure 9B). Other notable phosphopeptides identified were enriched in abnormal synaptic transmission (MP: 0003635; FDR: 0.0247; Top ranked constituents: Tbr1, Cacnb3, Stx7, Mapk8ip3, and Palm3), homeostasis/metabolism phenotype (MP: 0005376; FDR: 0.0044; Top ranked constituents: Tbr, Irs2, Prkra, Elmo1, Ccm2, Cacnb3, Gab1, Fabp3, Csde1, Adcy6, Stx7, Atg4c, Hmgcl, and Mapk8ip3), and abnormal learning/memory/conditioning phenotypes (MP: 0002063; FDR: 5.5 × 10^−4^; Top ranked constituents: Tbr, Cacnb3, Itsn1, Sipa1l1, and Pebp) (Figure 9C). Tbr (Rank: 14) and Cacnb3 (Rank: 24) were present in all three phenotypes and may be the link between LIB-induced cognitive decline driven by metabolic disruption and synaptic pathology (Figure 9D).

## 3. Discussion

### 3.1. General Overview

The findings presented in this study shed light on the intricate molecular networks underlying BINT and its impact on cognitive function, particularly in the context of genetic susceptibility to tauopathies. Through a comprehensive analysis integrating phosphoproteomics with behavior performance and network modeling techniques, several key insights have emerged.

First, the study revealed significant alterations in phosphopeptide profiles following LIB exposure, particularly in mice overexpressing the human tau P301L mutation. These alterations were distinct between LIB-exposed rTg4510 and nc mice, highlighting the interplay between genetic predisposition and environmental insult in shaping molecular responses to BINT. The observed hyperphosphorylation of tau at the pSer262 site in rTg4510 mice post-BINT suggests a potential mechanism linking blast exposure to tauopathy, which warrants further investigation.

The study utilized machine-learning-driven capabilities to predict cognitive outcomes based on phosphopeptide expression profiles. While predictions indicated reduced learning in LIB-exposed mice irrespective of genotype, actual learning performances revealed inconsistencies, particularly in rTg4510 mice where blunted responses to LIB were observed. This disparity unveils the complexity of the relationship between molecular changes and behavioral outcomes, highlighting the need for further refinement of predictive models.

Importantly, WpCNA identified distinct phosphopeptide modules associated with cognitive function, including modules linked to associative learning and mossy fiber pathways. These modules emerged as the most predictive of learning levels, highlighting their potential as biomarkers for cognitive impairment following BINT.

Furthermore, individual-level analysis identified specific phosphopeptides within these modules that conferred an increased risk for learning deficits following LIB exposure, particularly in rTg4510 mice. The identification of candidate phosphopeptides, such as Mapk8ip3 and Arhgap33, provides potential targets for future therapeutic interventions aimed at mitigating the cognitive sequelae of BINT, especially in genetically susceptible populations.

Group-level analysis using protein prioritization methods identified blast-relevant phosphopeptides, most notably Arhgap33, associated with abnormal behavioral and molecular phenotypes. These findings further unveil the multifaceted nature of BINT-induced learning deficits and provide insights into potential therapeutic targets for mitigating its adverse effects on cognitive function.

Taken together, this study elucidates the complex molecular mechanisms underlying BINT-induced cognitive decline, particularly in the context of genetic susceptibility to tauopathies. By integrating advanced analytical techniques, it offers valuable insights into potential therapeutic targets and biomarkers for assessing and mitigating the long-term consequences of BINT, thus advancing our understanding and management of this pressing public health concern.

### 3.2. Elevated Tau-ser262 Following LIB Exposure

As expected, most rTg4510 tau phosphopeptides were increased by mutant tau transgene expression, but surprisingly, we identified one LIB-induced tau phosphopeptide (SKIGSTENLK, Appendix A—MS^2^ significant phosphopeptides_rTg4510), which reflected increased ser262 phosphorylation in the second microtubule (MT) binding domain. Increased tau ser262 phosphorylation prevents MT binding and could indicate activation of one of the MARK family proteins. Although we did not find direct phosphosite evidence of increased activation of a MARK, studies have found that CamKIIa phosphorylation at the autoactivation site (Thr286) was significantly elevated in the LIB rTg4510 consistent with elevated Ca++ flux and glutamate-dependent excitotoxicity [52]. CamKIIa can directly or indirectly increase tau ser262 phosphorylation [53] and tau-dependent neurodegeneration which was reported to occur with axonal mitochondrial insufficiency [54], a feature at the intersection of tauopathy and post-TBI metabolic deficits.

### 3.3. WpCNA Module Eigenpeptides Predict and Correlate with Learning Behavior

Studies using this weighted co-expression network approach in the open-field blast setting have not been reported as of yet. Weighted co-expression network analysis has been used to predict prognosis in patients with colorectal adenocarcinoma and glioblastoma [55,56], to identify key genes related to HBV-associated hepatocellular carcinoma [57], and to classify individual humans into TBI or control groups based on WGCNA results [58]. Using this approach, we identified a key phosphopeptide module (turquoise module) that predicted learning index (Figure 7). Interestingly, functional annotation of the 129 phosphopeptides in this module revealed associative learning and myelin sheath to be the most enriched GO biological process and cellular component (Figure 6A). This is consistent with TBI and tauopathy-induced axonal damage reflected in individual phosphopeptide expression linked to disrupted axonal transport [59].

Our findings provide support that WpCNA is sufficient to identify groups of phosphopeptides, represented as co-expression modules, that correlate with, and can predict real learning trends in animals. The turquoise and black modules (described below in Section 3.4.1 and Section 3.4.2, respectively) displayed the greatest correlation strength with learning index and were clustered together based on their high intermodular topological overlap (Figure 4B). Support for these observed correlations with learning is that they were validated using MDS, which showed significant overlap between the molecular hubs of these two modules (Figure 5C). These modules displayed high predictive ability for learning levels. However, the overall ROC model accuracy and quality was most significant in the turquoise module.

### 3.4. WpCNA Individual-Level Analysis

This study identified individual-level phosphopeptidomic changes that may confer learning deficits following a single LIB exposure. The rTg4510 4 animal was found to have a globally distinct learning index, indicating that this mouse exhibited heightened vulnerability in tau-blast interactions, which was associated with more severe learning deficits (Figure 2C). Notably, in the rTg4510 BINT reduced absolute value of the module eigenpeptide, but lowered it disproportionately in this animal. This animal also displayed the lowest learning index for all 20 animals. Our blast experimental results suggest this animal experienced near-identical static and reflective overpressures as other animals and is not a justification for its distinct response (Appendix A). This suggests that not only are there significant group and transgene effects of LIB in mutant tau mice on phosphorylation (Figure 2A), but even with consistent pressures, mice within the rTg4510 BINT group show differential vulnerability to a single LIB exposure in learning defects and the phosphopeptidome.

#### 3.4.1. Turquoise Module

The turquoise-module-derived combined analysis across all the mice highlighted five increased phosphopeptides (Mapk8ip3, Gap43, Gab1, Arhgap33 and Slc1a2) and four decreased phosphopeptides (Otub1, Elmo1, CCM2 and Trip12) in the turquoise module for the rTg4510 4 animal relative to all other animals (Figure 8A). Mapk8ip3 (JIP-3) is a scaffold protein regulating JNK activation with rare missense mutations causing cognitive deficits [60] implicated in the regulation of axonal transport of dynein and kinesin-1 and bi-directional cargo [61,62], including lysosomes [63]. ROS-responsive ASK1 phosphorylates JIP-3 to enhance scaffolding interactions with SEK1/MKK4, MKK7, and downstream JNK3, promoting neurodegenerative JNK3 activation [64]. Consistent with a role in our model, TBI induces Rock1-dependent neurodegeneration [65] and phosphorylation at three sites in JIP-3:Ser^318^, Ser^368^, and Ser^369^ [66], resulting in JNK activation. Gap43 is well known as a primary JNK-targeted axonal phospho-protein [67] with elevations in AD CSF associated with tauopathy that predict progression [68]. Gab1 is also an adaptor protein with phospho-site-mediated coupling to SHP2 upstream of excitatory neuronal ERK activation and synaptic plasticity [69]; it is implicated in cognitive deficits in AD [69]. Arhgap33/Snx26 interacts with Sort1 to play a central role in axonal–synaptic TrkB trafficking [70] required for BDNF neuroprotective signaling. As a brain-enriched RhoGEF, it also interacts with rac>Cdc42 and control of dendritic spine formation and synaptogenesis [71]. Slc1a2 (Eaat2) is a glutamate transporter with a primary role in presynaptic and astrocytic glutamate reuptake and protection from excitotoxicity associated with TBI, tauopathy, and AD. PKCalpha is elevated by impact [72] and blast TBI [73] and phosphorylates Slc1a2/EAAT2 at Ser562/563 which induces glutamate excitotoxicity [74]. Otub1 is a Tau deubiquitinase in vitro and in vivo, involved in the formation of pathological tau, including small soluble oligomeric forms [75]. CK2 phosphorylation of cytosolic Otub1 at ser16 induces nuclear translocation [76]. ELMO1 is a Dock-regulated RacGEF that regulates Rac activation and control of actin assembly in dendritic spines [77], while NMDAR-mediated LTD induces KIF21B binding with Elmo1 and its translocation out of dendritic spines [78]. Known phospho-regulation is through Tyro3, Axl, and Mer receptor tyrosine kinases that phosphorylate Tyr18 and Tyr48 to promote RAC activation. TBI-induced Rock1 phosphorylation (for example of JIP-3) is inhibited by CCM2 [79] which acts as a scaffold for Rac1, actin, MEKK3/MAP3K3, and MKK3/MAP2K3 upstream of MAPKs and has multiple phosphorylation sites that bind major tau kinases including GSK3beta, CDK5, CK2, and Erk2. Finally, Trip12 is an E3 ubiquitin ligase with rare loss-of-function mutations implicated in cognitive deficits and autism [80]. Trip12 expression is elevated in human blood samples after controlled exposure to moderate blast in vivo [81]. Trip12 has 3 known phospho-sites and 76 interactors; however, its mechanistic role in LIB-induced cognitive deficits in rTg4510 remain unclear.

#### 3.4.2. Black Module

In the black module, Bcl2l13 was reduced in the rTg4510 4 animal relative to all other animals (Figure 8B). The brain deficiency of Bcl2l13 causes reduced mitophagy [82], while loss of an active p-Bcl2l13 predicts downstream reduction in mitophagy that could contribute to increased mitochondrial damage in tauopathy models and human tauopathies. Overall, our individual level phosphoproteomic analysis suggests plausible roles for multiple pathways potentially contributing to tau-related, BINT-induced cognitive deficits worthy of more detailed confirmation and follow-up exploration.

### 3.5. WpCNA Group-Level Analysis by Blast-Relevance Identified Phosphopeptides Related to Cognitive Decline, Synaptic Dysfunction, and Metabolic Disruption

Group-level analysis was performed to identify the turquoise and/or black module phosphopeptides that were most changed across five different blast vs. sham comparison groups (Figure 9B). Rank order statistics identified 40 blast-relevant phosphopeptides, 17 of which were related to abnormal learning and memory, synaptic dysfunction, and homeostasis and metabolic disruption phenotypes (Figure 9B). Among these, Otub1, Ccm2, Mapk8ip3, and Gab1 were also identified in the individual-level analysis to have high cumulative distance in the rTg4510 4 animal (Figure 8A). Notably, these phosphopeptides were all related to metabolic disruption in the group-level analysis (Figure 9C,D), though the literature supports their additional contribution to tauopathy, cognitive decline, and synaptic dysfunction [60,69,75]. Arhgap33 was not related to the selected phenotypes in Figure 9C,D; however, it is important to mention that this phosphopeptide was the top ranked (Figure 9B) and also identified through individual level analysis (Figure 8A). The most significant changes were in the rTg4510 blast vs. nc sham and rTg4510 blast vs. nc blast comparison groups (Figure 9B), indicating this phosphopeptide is associated with tau-blast interactions and possibly driving the distinct response in the rTg4510 4 animal. Arhgap33 interacts with sortilin cooperatively to facilitate trafficking of TrkB to synapses which is impaired in schizophrenia [70]. Functional knock-out of this protein in mice alters social behavior and cognition related to an autism-like phenotype [70]. These mice also have reduced dendritic spine density in the dentate gyrus and reduced firing frequency which have been linked with working memory deficits, learning deficits, habituation abnormalities, and anxiety [70]. This may be linked to the altered expression of phosphopeptides in the CA3 mossy fiber pathway network (Figure 6B; black module). Additionally, Arhgap33 displayed high diagnostic prediction of metabolic syndrome in patients with polycystic ovarian syndrome, which may be due to disrupted regulation of glucose transport, supporting its role in metabolic disruption [83]. However, this is the first study demonstrating its potential role in tau-related, BINT-induced cognitive decline.

### 3.6. Limitations

#### 3.6.1. Other Genetic Risk Factors Related to BINT-Induced Cognitive Decline

It is important to consider whether genetic markers other than tau play a role in BINT-induced cognitive decline. Dr. T. Wooten and colleagues (Boston VA and Tufts University) assessed neuropsychiatric function in 488 post-9/11 veterans with chronic blast exposure and discovered that those with a positive Apolipoprotein ε4 (Apo ε4) displayed significantly worse performance in the domain of memory than Apolipoprotein ε4 negative veterans [84]. Furthermore, Apoε4 positive veterans exposed to close-range (<10 m) blast overpressures were found to have significantly more white matter degradation than Apoε4-negative blast exposed veterans, thus unveiling a potential interaction between blast physics and AD genetic markers [85].

#### 3.6.2. Ongoing External Validation

Limitations of the study also include that only males were studied. The samples sizes allowed only statistical detection of medium effects. It is important to consider whether using biochemistry methods such as Western blot, enzyme-linked immunosorbent assay (ELISA), and/or immunoprecipitation as our ongoing studies. Due to the inherent limitations in omics-based biomarker detection from brain tissues, external validation in the circulating biofluids is required to confirm that a candidate biomarker’s altered expression is not a false positive.

#### 3.6.3. CognitionWall Platform

The use of the CognitionWall platform for assessing learning abilities in mice post-blast introduces inherent limitations. While this system provides a controlled and monitored environment, it may not fully represent the complexity of cognitive functions impacted by TBI. Additional behavioral paradigms and cognitive assessments should be considered to obtain a more comprehensive understanding of the cognitive sequelae following blast exposure.

#### 3.6.4. WpCNA Approach

WpCNA relies on correlation metrics for network construction, and its outcomes are influenced by the choice of parameters such as soft-thresholding power (β). While we utilized a β value that resulted in a scale-free network, the robustness of this choice may vary in different experimental settings. Readers should exercise caution in interpreting results and consider the sensitivity of the analysis to parameter adjustments. WpCNA assumes linear relationships between variables, and the biological processes underpinning phosphoproteome changes and learning abilities may exhibit non-linear dynamics. This assumption limits the ability of WpCNA to capture intricate non-linear associations within the data. Integrating complementary analytical approaches may provide a more holistic understanding of the complex relationships involved.

### 3.7. Future Studies

Future studies should aim to address the limitations identified in this investigation to advance our understanding of BINT and its impact on cognitive function. Firstly, further exploration of genetic risk factors beyond tau, such as Apo ε4, is warranted to elucidate their role in BINT-induced cognitive decline. This could involve large-scale cohort studies to validate the interaction between blast physics and AD genetic markers. Ongoing external validation of identified disease-modified phosphopeptides is essential to confirm their significance as biomarkers for poor cognitive outcomes. While the CognitionWall platform provides valuable insights into learning abilities post-blast, additional behavioral parameters and cognitive assessments should be considered to capture the full spectrum of cognitive impairments following BINT. Additionally, we will pursue the study of sex and age impacts on behavioral and molecular responses to BINT. The limitations of the WpCNA approach should be addressed by integrating complementary analytical methods. Future studies could explore non-linear associations within the data and assess the sensitivity of the analysis to parameter adjustments, thus providing a more comprehensive understanding of the complex molecular networks underlying BINT-induced cognitive decline.

## 4. Materials and Methods

### 4.1. Open-Field Primary LIB in Mice

All animal experiments were conducted in accordance with the approved protocols for the Care and Use of Laboratory Animals and the Animal Research: Reporting of In Vivo Experiments (ARRIVE) guidelines at the University of Missouri. The mice were housed in standard home-cages with bedding, maintained on a 12-h light/dark cycle, and provided with ad libitum access to food and water throughout the study period. The open-field LIB exposures were performed at the Missouri University of Science & Technology, following established procedures as previously reported [4,11,13,14,40,43,44,86]. The study included a total of 20 male mice: 10 rTg4510 (RRID: IMSR_JAX:024854) [Tg(CaMKIIa-tTA)/Mmay and Fgf14/Tg(tetO-MAPT·P301L)4510 mice] and 10 non-carrier (RRID: IMSR_JAX:019019) mice in the same C57BL/6J background recommend by and purchased from Jackson Laboratory (Bar Harbor, ME, USA), both aged 2 months. The mice were randomly divided into four experimental groups: five rTg4510 mice and six non-carrier mice exposed to LIB; five rTg4510 mice and four non-carrier mice were subjected to sham procedures. Mice in the sham control groups underwent identical anesthesia procedures without LIB exposure. Prior to the LIB exposure, the mice were anesthetized using an intraperitoneal injection of a ketamine/xylazine mixture, with a dosage of 8 μL/g bodyweight (12.5 mg/mL ketamine and 0.625 mg/mL xylazine). To immobilize the mice during LIB exposure, they were placed in upright positions in 3D-printed chairs made of carbon reinforced nylon (Nylon X; Matterhackers, Lake Forest, CA, USA), as previously described [40]. Elastic mesh bands were employed to restrict head and body movements. The chairs were designed with a streamlined structure to minimize shock impingement and reflection while providing adequate thoracic support and preventing head and trunk movements during primary LIB exposure. The mouse holders within the 3D-printed chairs were positioned 3 m away from the detonation site of a 350 g C4 explosive generating a magnitude of 46.6 kPa (6.67 psi) peak overpressure, a maximal impulse of 60 kPa·ms (8.70 psi·ms) (Figure 1, #1) [40]. Following LIB exposure, the mice were returned to their original cages. After a recovery period from anesthesia, mice were closely monitored for at least 15–30 min, during which they were allowed access to food and water ad libitum to facilitate their post-exposure recovery.

### 4.2. Automated Assessments of Learning Ability in a Home-Cage Environment

Learning index in the four animal groups was measured using the PhenoTyper home-cages (Model 3000; Noldus Information Technology, Wageningen, The Netherlands) and CognitionWall system (Noldus Information Technology, The Netherlands), as previously described (Figure 1, #2) [87,88,89]. Prior to conducting the CognitionWall assessments, all mice were acclimated to the PhenoTyper home-cages. They were placed in the PhenoTypers for three days to familiarize them with the home-cage environment. At 3:00 p.m., regular food chows were removed, and at 3:45 p.m., the CognitionWall devices were inserted into the PhenoTypers at the back-left corner. The CognitionWall had three entrances (left, middle, and right) placed in front of the food dispenser. Mice were required to enter the CognitionWall through the left entrance to receive a reward of one food pellet every fifth time of the correct entry, following a Fixed Ratio 5 schedule (FR5 schedule) during the experiment. Mouse behavior was automatically recorded using a 24/7, infrared-sensitive video-based observation system located on the top unit of the PhenoTypers. The mice were continuously monitored for 48 h. All animal tracking data were acquired through EthoVision XT software v14 (RRID: SCR_000441; Noldus Information Technology, The Netherlands) and sampled at a rate of 15 fps. Learning index for day 1 and 2 of discrimination learning was calculated as previously described [88]; however, to evaluate the growth in learning over the course of 48 h, we subtracted learning index day 1 from learning index day 2 (Equation (3)).
(3)$$\mathrm{Learning}\;\mathrm{Index}\;\left(\mathrm{Day}2-\mathrm{Day}1\right)=\frac{\left({\mathrm{CE}}_{D2}-{\mathrm{IE}}_{D2}\right)}{\left({\mathrm{TE}}_{D2}\right)}-\frac{\left({\mathrm{CE}}_{D1}-{\mathrm{IE}}_{D1}\right)}{\left({\mathrm{TE}}_{D1}\right)},$$
where CE = correct entries; IE = incorrect entries; TE = total entries; D2 = day 2; D1 = Day 1. For logistic regression only, the learning index continuous variable was transformed into a categorical learning “scale” by plotting a normal distribution of the learning index where each learning level was determined by the number of standard deviations to the left or right of the mean.

### 4.3. Tissue Collection and Protein Extraction

Mice were sacrificed at 40 days post-injury (dpi). Brains were dissected and processed as described previously [43,90]. The brain tissue of the frontal cortex was collected for analysis (Figure 1, #3). Briefly, sample lysis buffer (2% sodium dodecyl sulfate [SDS], 0.5 M tetraethylammonium bicarbonate [TEAB], pH 8.5 and protease inhibitor cocktail) was added to each tissue specimen. Specimens were homogenized by Glas-Col stringer 099C K43 (Glas-Col LLC, Terre Haute, IN, USA) and centrifuged at 17,000× *g* for 20 min at 4 °C. The supernatant was collected and then precipitated by cold acetone.

### 4.4. Protein Digestion for 4-D Tandem Mass Spectrometry

Each protein sample (n = 20) was centrifuged and washed with 80% acetone twice, and protein pellets were then resuspended with 6 M urea, 2 M thiourea, and 100 mM ammonium bicarbonate. Protein was quantified using the Pierce 660 nm Protein Assay method following the microplate measure instructions in the manual. A total of 700 µg of protein from each sample was reduced and alkylated. Then trypsin (ratio 1:50 trypsin: protein, w/w) was added for digestion at 37 °C overnight. Digested peptides were purified by Pierce C18 tips. A total of 5% digested peptides were used for total proteome analysis. The remaining 95% peptides were subjected to phosphopeptide enrichment by High-Select™ TiO_2_ Phosphopeptide Enrichment Kit (Cat #: A32993; Thermo Fisher Scientific, Waltham, MA, USA) (Figure 1, #4). The pooled samples were fractionated by Pierce High pH Reversed-Phase Peptide Fractionation Kit according to the manual instructions. For phosphorylated peptides, five fractions were collected (5%, 7.5%, 10%, 15% and 50% ACN). One microliter of suspended peptide was separated on a C18 column (20 cm × 75 µm × 1.7 µm) with a step gradient of acetonitrile at 300 nL/min. Initial conditions were 1% B (A: 0.1% formic acid in water, B: 99.9% acetonitrile, 0.1% formic acid), followed by 34 min ramp to 17% B, and then 17% to 25% B over 8 min, 25–37% B over 6 min, gradient of 37% B to 80% B over 4 min, hold at 80% B for 4.5 min, back to 1% in 0.5 min, hold at 1% B for 3 min. Total run time was 60 min. For data-dependent acquisition (DDA), MS data were collected in positive-ion data-dependent PASEF mode over an m/z range of 100 to 1700 Da and ion-mobility range of 0.57 to 1.6 1/k_0_. During MS/MS data collection, each TIMS cycle included 1 MS + an average of 10 PASEF MS/MS scans, the total time per cycle is 1.16 s. For DIA MS data were collected in positive-ion data-independent PASEF mode over an m/z range of 400 to 1200 Da and an ion-mobility (IM) range of 0.57 to 1.47 1/k_0_. A total of 64 DIA-PASEF windows were used (25 m/z steps and 0.18 IM steps) with two collision energies based on IM.

### 4.5. Spectral Library Generation, Phosphopeptide Identification, and Raw Data Processing

Five fractions (5%, 7.5%, 10%, 15% and 50% ACN) analyzed by DDA acquisition were used for library build-up. The library was built using DDA data by Pulsar in Spectronaut, filtered with PSM/peptide/protein FDR 1%. The protein database of Uniprot mouse protein database of up00000589 (17,058 items) and human tau protein sequence was employed. The following parameters were used: trypsin-specific digestion, 2 missed cleavages, carbamidomethylation on Cys as fixed modification, acetylation on protein N-term, and oxidation (M) as variable modifications. (For phosphopetides library, phospho(STY) was also added to the variable modification. DIA data were searched using Spectronaut with the spectra library generated by Pulsar. The precursor/protein Q-value cutoff was 0.01. Quantitation type of MS^2^ area was chosen. Data filtering was set to Q-value sparse (for phosphopetides analysis, PTM localization probability cutoff was set to 0.75). The data were normalized using the sum-normalization method. Raw output data including a column for phosphopeptide identifiers and 20 columns for phosphopeptide log_2_transformed intensities across all 20 samples were used as input for WpCNA. Prior to WpCNA, behavior prediction was performed based on phosphoprotein expression using IPA machine learning capabilities (Figure 1, #5).

### 4.6. Processing of Expression (Phosphopeptidome) and Functional Outcome (Learning Index) Datasets for WpCNA

WpCNA was conducted in accordance with previously described methods [47]. Raw MS^2^ expression data (log_2_transformed phosphopeptide data) and learning index data frames were imported into RStudio (RRID: SCR_000432; version: 2023.12.1+402) for WpCNA (RRID: SCR_003302) (Figure 1, #6). Prior to module detection, a cluster tree dendrogram using hierarchical clustering methods was created to conceptualize the relationship between blast conditions and different genotypes with learning index, highlighting the functional outcome differences within and between all 4 animal groups. Marginal Pearson correlation was adopted as a standardized gene screening method, to screen phosphopeptides that do not correlate with learning index across all 20 animals. Phosphopeptides were removed from the expression data frame if Pearson coefficient was I ≤ 0.5.

### 4.7. Adjacency Matrix Construction and Validation of Scale-Free Topology

An adjacency matrix (A) or connection matrix can be defined as applying a soft-thresholding power parameter (β ≥ 1) to the absolute value of the pairwise correlation (similarity matrix, s_ij_) between phosphopeptides (x_i_, x_j_) (Equations (4) and (5)). This has proven to yield more robust results than unweighted networks [50]. An adjacency matrix converts the original expression matrix [x_1_, x_2_ … x_n_] to a symmetrical square matrix (x_n_ · x_n_) with values 0 and 1 indicating no connection or connection between two phosphopeptides. This will be referred to as the “co-expression” matrix
(4)$$s_{\mathrm{ij}}=\left|\mathrm{cor}\left(I,x_{j}\right)\right|$$
(5)$$A_{\mathrm{ij}}={\left(s_{\mathrm{ij}}\right)}^{\beta }$$
where β is chosen according to scale-free topology criterion; if data meets scale free topology criterion, the power law is achieved, indicating that the vast majority of nodes (phosphopeptides) have very few connections, while a few important nodes (hubs) have a huge number of connections as is the case in most biological samples [50]. To determine whether the phosphopeptide expression data frame meets the scale-free topology criterion, a new variable, k (network connectivity), was used in place of A_ij_ and s_ij_. Applying a logarithmic transformation to Equation (5), one can see that the adjacency network connectivity is directly proportional to the similarity network connectivity (Equation (6)). Scale-free topology is achieved where plotting Log(A_ij_) vs. Log(s_ij_) or Log(A_k_) vs. Log(s_k_) yields a straight line. Importantly, β = 8 was used for this proteomic study because it was the only value for β where scale-free topology was achieved.
(6)$$\mathrm{Log}\left(A_{k}\right)=\beta \left(\mathrm{Log}\left(s_{k}\right)\right)$$

### 4.8. Construction of Topological Overlap Matrix (TOM)

While the adjacency matrix considers two pairs of phosphopeptides in isolation to assess weighted similarity, the topological overlap matrix considers two pairs of phosphopeptides in relation to all other phosphopeptides in the dataset. If two nodes (x_i_ and x_j_) are said to have topological overlap, they are connected to the same group of nodes (x_u_). The TOMdist function in r computes the topological overlap of the x_n_ · x_n_ square adjacency matrix using Equation (7).
(7)$${\mathrm{TOM}}_{\mathrm{ij}}\left(A\right)=\frac{\sum_{u\neq i,j}a_{\mathrm{iu}}a_{\mathrm{uj}}+a_{\mathrm{ij}}}{\min\left({\sum^{\;}}_{u\neq i}a_{\mathrm{iu}},{\sum^{\;}}_{u\neq j}a_{\mathrm{ju}}\right)+1-a_{\mathrm{ij}}}$$
where the topological overlap between nodes i and j is a function of the adjacency between nodes i and j (a_ij_), adjacency between node i and another node u (a_iu_), and the adjacency between node j and node u (a_ju_). TOM is a measure of weighted interconnectedness between two phosphopeptides and similar to adjacency, in that it also outputs values of 0 and 1. For WpCNA, TOM is referred to as the TOM similarity.

### 4.9. Detection of Phosphopeptide Module Membership

Modules in the weighted gene co-expression network analysis (WGCNA) create new variables (e.g., eigengene, eigenpeptide) within the first principal component that represents the expression of a subset of interconnected nodes [50]. Hierarchical clustering was used to cluster phosphopeptides by topological overlap. Phosphopeptide clusters were grouped into modules based on the cutting method chosen and the tree cut height. Three cluster tree cutting methods such as colorstaticTOM, colorDynamicTOM, and colorDynamicHybridTOM were compared. Phosphopeptides that do not fit into a distinct module are grouped into the grey module. Phosphopeptide expression profiles within a given module can be summarized by the module eigenpeptide which can be utilized in linear-mixed models and Bayesian networks [50]. Phosphopeptide module members were characterized using a STRING (RRID: SCR_005223; version: 12.0).

### 4.10. Relating Module Eigenpeptide to Learning Index

Peptide screening was performed to correlate phosphopeptide expression with learning index for all 20 animals by computing the marginal Pearson correlation and *p* value between phosphopeptide Log_2_ transformed intensities and learning indices. To identify modules that are related to learning index, WpCNA performs a standard Pearson correlation between each log_2_-transformed phosphopeptide intensity for each module and the learning level across all 20 animals which we refer to as the peptide significance, identical to gene significance; the module significance is computed by taking the average of all of its peptides significances (Figure 1, #6). Network screening was then performed to identify module phosphopeptides with high individual weighted and unweighted Pearson correlation with learning index (Figure 1, #6). The weighted Pearson correlation factors in peptide significance and module membership features of each phosphopeptide for each module. To determine whether module eigenpeptides (independent variables) can predict a learning index, a multiple logistic regression was performed (Figure 1, #6). And a plot of the receiver-operating characteristics (ROC), which predicts outcome (or dependent variable, e.g., learning index) based on the value of the independent variables (module eigenpeptides for each module), was constructed to visualize model fitness and the area under the curve (AUC); AUC *p* value and goodness-of-fit were computed to identify modules that were predictive of learning index.

### 4.11. Statistical Methods

#### 4.11.1. Behavioral Analysis

The discriminatory learning index was computed using Equation (3). Group-level statistical comparison tests for leaning index differences were assessed using a One-Way ANOVA (RRID: SCR_002798; GraphPad Prism; version: 10).

#### 4.11.2. MS^2^ Quantitative Proteomics

DIA-PASEF MS^2^ data was searched using Spectronaut (version: 15.6) with the spectra library generated by Pulsar. The precursor/protein Qvalue cutoff was 0.01. Quantitation type of MS^2^ area was chosen. Data filtering was set to Qvalue sparse (for phosphopetide analysis, PTM localization probability cutoff was set to 0.75). Raw output data including a column for phosphopeptide identifiers and 20 columns for phosphopeptide log_2_transformed intensities across all 20 samples were used as input for WpCNA. Group level LIB phosphoproteomic MS^2^ intensity differences were assessed by performing a two-tailed Student’s *t*-test between blast and sham animal groups for both genotypes. Fold change was calculated by dividing LIB-exposed non-carrier or rTg4510 Log_2_transformed intensity by unexposed sham non-carriers or rTg4510 mice, respectively.

#### 4.11.3. ROC Analysis

To assess whether module eigenpeptides (i.e., summary values of each animal’s phosphopeptide expression in that module) can sufficiently discriminate between non-carrier and rTg4510 LIB-exposed and unexposed sham controls’ learning level, we performed ROC analysis under nonparametric assumption. Independent and dependent variables were set to the module eigenpeptides for each animal (n = 20) and their learning level, respectively. The positive “actual state” was set to 4.0, the mean learning level (Figure 2A). Evaluation of the models’ strength was assessed by observing the area under the curve (AUC), where if AUC = 0.5, the model performs no better than at random. Overall model quality was assessed using the Kolmogorov-Smirnov (K-S) metric. The K-S statistic is often used to quantify the separation between the true positive rate (sensitivity) and the false positive rate (1 − specificity) across different classification thresholds [91]. The K-S statistic is calculated by identifying the maximum vertical distance between the ROC curve and the diagonal line (representing random chance). The combination of the Kolmogorov-Smirnov (K-S) test and the AUC/ROC curve provides a more detailed evaluation of model performance [91]. These analyses were performed in SPSS (RRID: SCR_002865; version 29.0.2.0).

## 5. Conclusions

In conclusion, our study demonstrates that the WpCNA module eigenpeptides derived from the phosphoproteomic landscape effectively predict learning behavior in rTg4510 mice, a model exhibiting accelerated tau pathology. The identified turquoise and black modules, characterized by increased phosphopeptides associated with learning deficits, were validated through ROC modeling, affirming WpCNA’s ability to discern phosphopeptide patterns indicative of cognitive decline. Notably, our investigation into the molecular mechanisms following LIB exposure revealed elevated phosphorylation of tau at Ser262, coupled with increased CamKIIa phosphorylation, indicating a potential link between BINT, tau aggregation, and synaptic excitotoxicity. The WpCNA analysis further uncovered key phosphopeptides including Arhgap33 through individual and group level analysis that were associated with cognitive decline and tauopathy, synaptic dysfunction, and metabolic disruption, shedding light on potential therapeutic targets guided at associative learning and CA3 mossy fiber pathways. Our findings reveal the potential of phosphoproteomic approaches, particularly WpCNA, in unraveling intricate molecular networks underlying learning deficits in the context of tauopathy and BINT. Further validation and exploration of these identified pathways hold promise for advancing our understanding of the interplay between ADRD genetic risk factors, phosphoproteomic network alterations, and cognitive outcomes following BINT.

## Data Availability

The data presented in this study, including the code used for WpCNA, are contained within this manuscript in Appendix A. If additional data are requested, the authors will provide it.

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
