# Peer review of "The Chronic Effects of a Single Low-Intensity Blast Exposure on Phosphoproteome Networks and Cognitive Function Influenced by Mutant Tau Overexpression"

_ijms, 2024, doi:10.3390/ijms25063338_

Round 1

Reviewer 1 Report

Comments and Suggestions for Authors

The current manuscript by Jackson et al. seeks to elucidate the impact of blast-induced neurotrauma on cognition in genetically heterogenous mice linked to frontotemporal dementia. The goals of the study are relevant to the readers of IJMS and the studies are timely in their relevance to the field. The data is compelling and may be useful others within the field. There are issues with some aspects of the experimental design that should be addressed, however. These are addressed below.

1.    The authors utilized an inducible transgenic model for the study of FTD without the proper controls. This is an inducible transgenic line that should have utilized non-induced littermates as controls. The controls for the study are sourced C57Bl/6J mice from Jackson laboratories. The control mice utilized for the study do not even match the strain that the rTg4510 mice are on (FVB/N). The authors use the phrase “genetically heterogeneous” to describe the subjects used during their studies. This makes it difficult to impossible to discern what effects are perhaps strain specific in nature. Much of the data is compelling however and adds to the literature, but the lack of this as a limitation to the study should certainly be noted in the discussion. This is a much larger limitation to the study than any sex and or age differences that may be present and these are noted within the discussion.

2.    The emphasis on Blast rTg4510 4 animal is very odd. The reviewer understands that this subject clearly had severely slowed learning, but it is certainly an outlier within their groups. It is odd to have so much focus on a singular animal that is clearly an outlier compared to the others of their groups, even within the same experimental group and genetic background.

3.    On this same point, there are no acute measures to ensure that Blast rTg4510 4 animal did not receive a “worse” injury than other subjects directly driving the enhances response. Were all the subjects exposed to the same C4 charge or were these experiments independently conducted? Additionally, if these were independent experiments were the charges and associated overpressures and impulses measured independently to ensure consistency across the BINT groups?

4.    A highpoint of the study is that a similar framework may be used by others in looking at phosphopeptides across other models for TBI and their relation to learning. This increases the impact of the current study greatly.

Reviewer 2 Report

Comments and Suggestions for Authors

Appreciate the authors on the efforts to understand the mechanism involved in blast exposure and cognitive decline. The findings are beneficial in the neurotrauma field. However, the structure presented is confusing for the reader. For example, the authors state their first objective in the second paragraph (1.2), then continue their interest in phosphorylated tau in clinical and preclinical studies. The structure should be clear with a focus on the main outcome (phosphoproteomic?) and followed by the secondary outcome (cognition?). 

The results section is too descriptive, not just the findings. the authors should summarize the findings so that the reader can understand. The authors should limit any discussion to the literature (also no citation) in the result. If you do so, citations should be included. 

Same thing for your discussion. Which part of your study do you want the readers to focus on? I might be naive.  You have listed many limitations for your paper and it is clear that the authors can avoid them (including females, older age mice....). I wonder why you chose these approaches during your experimental design when you clearly listed them as your limitations.   

Again, I am lost with the materials and methods section. The opening paragraph is the objective of the study?  Please check it again.     

Overall, the writing structure of this manuscript should be improved for clarity. 

Comments on the Quality of English Language

The presentation of your manuscript should be improve

Round 2

Reviewer 1 Report

Comments and Suggestions for Authors

The authors have adequately addressed the previous critiques and should be commended for their work that will add greatly to the blast field. 

Author Response

"Reviewer's comments have been fully addressed."

Reviewer 2 Report

Comments and Suggestions for Authors

Thank you for the responses. I have a few additional comments.

Lines 143-165: This paragraph appears redundant with the method sections you described. I would recommend a clear focus on the objective of the study rather than summarizing the methods section.

Lines 754-763: I am curious why this paragraph is in the limitations section when there's no argument with your study.

Lines 764-770: The sentences are redundant; please clarify.

Lines 1050-1068: This might be an error; the conclusion shouldn't follow the methods section.

Comments on the Quality of English Language

NA
